# Anti-Epileptogenic Effects of Antiepileptic Drugs

**DOI:** 10.3390/ijms21072340

**Published:** 2020-03-28

**Authors:** Barbara Miziak, Agnieszka Konarzewska, Marzena Ułamek-Kozioł, Monika Dudra-Jastrzębska, Ryszard Pluta, Stanisław J. Czuczwar

**Affiliations:** 1Department of Pathophysiology, Medical University of Lublin, 20-090 Lublin, Poland; barbara.miziak@umlub.pl (B.M.); agnieszka.konarzewska@umlub.pl (A.K.); monika.dudra-jastrzebska@umlub.pl (M.D.-J.); 2Laboratory of Ischemic and Neurodegenerative Brain Research, Mossakowski Medical Research Centre, Polish Academy of Sciences, 02-106 Warsaw, Poland; mulamek@imdik.pan.pl

**Keywords:** epilepsy, epileptogenesis, status epilepticus, neuroprotection, antiepileptic drugs, seizures

## Abstract

Generally, the prevalence of epilepsy does not exceed 0.9% of the population and approximately 70% of epilepsy patients may be adequately controlled with antiepileptic drugs (AEDs). Moreover, status epilepticus (SE) or even a single seizure may produce neurodegeneration within the brain and SE has been recognized as one of acute brain insults leading to acquired epilepsy via the process of epileptogenesis. Two questions thus arise: (1) Are AEDs able to inhibit SE-induced neurodegeneration? and (2) if so, can a probable neuroprotective potential of particular AEDs stop epileptogenesis? An affirmative answer to the second question would practically point to the preventive potential of a given neuroprotective AED following acute brain insults. The available experimental data indicate that diazepam (at low and high doses), gabapentin, pregabalin, topiramate and valproate exhibited potent or moderate neuroprotective effects in diverse models of SE in rats. However, only diazepam (at high doses), gabapentin and pregabalin exerted some protective activity against acquired epilepsy (spontaneous seizures). As regards valproate, its effects on spontaneous seizures were equivocal. With isobolography, some supra-additive combinations of AEDs have been delineated against experimental seizures. One of such combinations, levetiracetam + topiramate proved highly synergistic in two models of seizures and this particular combination significantly inhibited epileptogenesis in rats following status SE. Importantly, no neuroprotection was evident. It may be strikingly concluded that there is no correlation between neuroprotection and antiepileptogenesis. Probably, preclinically verified combinations of AEDs may be considered for an anti-epileptogenic therapy.

## 1. Antiepileptic Drugs—Mechanisms of Action

Bromides actually started pharmacotherapy of epilepsy more than 150 years ago but their therapeutic index was quite low [1,2]. In 1912, phenobarbital was introduced as an effective antiepileptic drug (AED) and it is actually the only antiepileptic whose discovery was not associated with animal seizure tests. Subsequently, more and more antiepileptics appeared on the market—phenytoin, carbamazepine and valproate followed by newer and, eventually, the newest AEDs [1,2]. AEDs share multiple mechanisms of action (topiramate, valproate and others) or exert just one mechanism, for instance tiagabine or vigabatrin. Benzodiazepines, carbamazepine, ethosuximide, phenobarbital, phenytoin and valproate belong to conventional (or the first generation) antiepileptics whilst gabapentin, lamotrigine, levetiracetam, oxcarbazepine, pregabalin, tiagabine, topiramate or vigabatrin may be examples of the second generation. The newest AEDs (or the third generation) for instance are brivaracetam, eslicarbazepine acetate, lacosamide and perampanel [3]. The crucial mechanisms of action of AEDs comprise three main targets: blockade of voltage-operated sodium or calcium channels, potentiation of GABA-mediated events and reduction of excitatory events associated with glutamate. The first mechanism is shared by carbamazepine, eslicarbazepine, gabapentin, lamotrigine, oxcarbazepine, pregabalin and phenytoin. GABA-mediated inhibition is enhanced by benzodiazepines, phenobarbital, topiramate, tiagabine, valproate and vigabatrin whilst inhibition of glutamate-mediated excitation is mediated by lamotrigine, perampanel and topiramate [2,3,4]. Some AEDs possess very particular mechanisms of action—levetiracetam or brivaracetam are ligands of synaptic vesicle protein (SV2A) and lacosamide, in contrast to other sodium channel inhibitors (carbamazepine or phenytoin which increase fast inactivation of sodium channels), is an accelerator of the slow inactivation of sodium channels [1,2,3,4]. Very recently, neural networks alterations associated with levetiracetam treatment have been recently described with the use of functional magnetic resonance imaging. It was evident that levetiracetam produced evident changes in the functional connectivity patterns of the cognitive networks in patients with temporal lobe epilepsy [5].

The authors assumed the literature search in this review for the time frame from 1995 until March 2020. Only English language articles were considered in PUBMED databases and the search areas included: mechanisms of action of AEDs, mechanisms of epileptogenesis, AEDs and epileptogenesis, inhibition of epileptogenesis by non-AEDs and other agents and clinical data on inhibition of epileptogenesis. A limited number of particularly important earlier publications was also considered.

## 2. Epileptogenesis

Usually, after several weeks following status epilepticus (SE), spontaneous seizures are generated indicating that the normal animal brain has been transformed into an epileptic one following SE-initiated epileptogenesis. It is thus clear that epileptogenesis is usually a long-term process converting the normal brain into one capable of generating seizure activity in patients after brain insults—head traumas, stroke or already mentioned SE [6]. Nevertheless, this process is not only associated with the silent period after SE and may be continued after the onset of seizures. Moreover, it can further exacerbate the frequency of spontaneous recurrent convulsions [7].

Three separate stages of epileptogenesis may be distinguished: initial insult (for example SE), latent period without seizure activity and finally, chronic epilepsy with recurrent seizure activity [8]. There are apparently acute (from hours to weeks, including transcriptional events, neurodegeneration and activation of inflammatory pathways) and chronic processes (mainly from weeks to months, embracing neurogenesis, mossy fiber sprouting, neuronal circuit reorganization and gliosis) which lead to and sustain epileptogenesis [8].

As already mentioned above, one of the probable mechanisms of epileptogenesis is hippocampal axonal sprouting leading to the formation of novel excitatory neuronal circuits. This particular mechanism is initiated between 1 and 2 weeks following kainate-induced SE and may persist up to 10 weeks or longer [7]. Hippocampal mossy fibers are actually axons of dentate granule cells projecting to the CA3 hippocampal area and in physiological conditions usually are not present in the dentate inner molecular layer [9]. In animals surviving SE but also in patients with mesial temporal epilepsy, mossy fibers show extensive branching and the branching collaterals project to the molecular layer forming excitatory synapses to external granule cell dendrites. There are studies indicating that the mossy fiber sprouting is positively correlated with seizure aggravation and possibly results from the reorganization of the dentate circuits [9]. Apart from mossy fiber sprouting, aberrant location of dentate granule cells may contribute to epileptogenesis. Eventually, dispersed granule cells produce a granule cell layer much wider than in healthy subjects and after all, many of them have aberrant location in the dentate hilus (ectopic granule cells). In healthy subjects this location is possible but the density of hilar granule cells is much lower. Some of ectopic granule neurons receive excitatory connections from regular granule cells and from pyramidal cells located in the hippocampal area CA3. Furthermore, ectopic granule neurons send out their axons both to the molecular layer and CA3 area. The excitatory circuits formed as indicated above are evidently responsible for hippocampal hyperexcitability and progressing epileptogenesis [9].

Gliosis has been found in the chronic phase of epilepsy and astrocytes have been documented to release a number of neurotransmitters and modulators—for instance, glutamate or adenosine. After all, astrocytes via releasing glutamate may participate in the induction of epileptiform discharges and have been ascribed to the pathogenesis of febrile seizures and temporal lobe epilepsy [8].

After all, epileptogenesis has been linked with distinct changes in the expression of more than 100 genes associated with a control of various signaling pathways. Evidences are available for transforming growth factor β (TGF-β), insulin-like growth factor 1 (IGF-1), p38MAPK, mTOR and many others [8].

## 3. Do Antiepileptic Drugs Inhibit Epileptogenesis?

As shown above, AEDs apart from their substantial therapeutic effect (prevention of seizure activity) achieved by various mechanisms, may also possess a neuroprotective potential, reflected by a clear cut protection of vulnerable brain structures against diffuse seizure-related brain damage. Neuroprotective effects may be achieved by various mechanisms, such as inhibition of sodium or calcium channels, blockade of glutamate receptors, enhancement of GABA-mediated inhibition, antioxidation, inhibition of apoptopic cascade, modulation of neurotrophic factors and many others [10]. Although AEDs share some of them, their neuroprotective potential may differ. All available AEDs may be divided into two groups—one characterized by distinct neuroprotective activity (benzodiazepines, lamotrigine, levetiracetam, phenobarbital, topiramate, valproate, vigabatrin) and the others sharing no such potential, as for instance carbamazepine or phenytoin against experimental seizure models [10,11,12,13]. Noteworthy, neurodegeneration may be considered as one of the crucial mechanisms involved in the process of epileptogenesis [14] so a pivotal question emerges whether preferably AEDs with the neuroprotective potential can actually stop or at least delay epileptogenesis. If this is the case then these AEDs could be regarded as agents able to modify the course and/or progress of epilepsy.

Seizure-related brain damage may be easily evaluated in models of SE either elicited by chemical means (peripheral or central administration of convulsive substances—kainate, pentylenetetrazol or pilocarpine) or electrical stimulation of vulnerable brain structures (hippocampus or amygdala). AEDs documented to fully inhibit or at least diminish the severity of spontaneous convulsions could be regarded as anti-epileptogenic.

In the model of SE produced by electrical stimulation of the perforant path in rats, tiagabine was administered subchronically via osmotic Alzet pumps (50 mg/kg/day) and the AED was started three days before the induction of SE [15]. Apparently, tiagabine reduced the seizure severity (duration and number of partial seizures were diminished whilst generalized seizures were completely blocked) during SE and, when evaluated 2 weeks after SE, exerted a clear cut neuroprotective effect in hippocampal pyramidal cells located in the CA1 and CA3 areas. Neuroprotection was also recorded in extrahippocampal areas and its extent in the hilus of the dentate gyrus was only moderate. The Morris water-maze test was also carried out two weeks following SE and tiagabine-administered animals evidently performed better. Spontaneous seizure activity was not measured in this study [15]. Another study, in the same model of SE in rats, evaluated carbamazepine and lamotrigine, taking as the endpoints neuroprotection and memory deficit in the water-maze test. Again, spontaneous seizures were not considered [16]. Administration of AEDs was initiated either 3 days before SE (pre-drug groups) or 1 h after SE (60 min of electrical stimulation of the perforant path; post-drug groups). Both antiepileptics were injected twice daily, carbamazepine at 30 mg/kg and lamotrigine at 12.5 mg/kg—the behavioral and histological procedures being brought about after 2 weeks of drug administration. Carbamazepine in both administration procedures was devoid of any neuroprotective activity in the dentate hilus, CA3 area and piriform cortex. However, neurodegeneration studied in the pre- and post-lamotrigine groups was mild (in CA3a,b pyramidal neurons and hilus) and comparable to that found in the control (without SE) group. Additionally, the CA3c area was also spared in pre-lamotrigine group. Regarding memory impairment, this was shown to be the case in all animals [16].

One of the first studies evaluating spontaneous convulsions involved diazepam given in a high dose of 20 mg/kg, 2 or 3 h after the onset of SE produced by electric stimulation of the rat amygdala [17]. Diazepam (given 2 h after) afforded a significant neuroprotection and considerably reduced the number of rats exhibiting spontaneous seizures as compared with the respective control group. When given 3 h after, the AED proved much weaker as a neuroprotective and anti-epileptogenic agent. Diazepam was also assessed at a much lower dose of 2.5 mg/kg, administered twice on the day of SE in rats induced by lithium/pilocarpine. In these circumstances, diazepam exhibited neither neuroprotective nor anti-epileptogenic potential [18]. The same research group evaluated the influence of topiramate, in doses of 10, 30 or 60 mg/kg, on the outcome of lithium/pilocarpine SE in rats. The AED was administered at 1 and then 10 h of SE and the injections twice daily followed for the next 6 days. Evidently, topiramate proved neuroprotective against SE-induced neurodegeneration in the hippocampus. In the dose range of 10–60 mg/kg, topiramate protected 24%-30% of neurons in the layer CA1 whilst only at 30 mg/kg neuroprotective effect was found in layer CA3b. In spite of the hippocampal neuroprotection, topiramate reduced neither the latency nor frequency of spontaneous seizures [18]. In the same model of SE, topiramate (10–60 mg/kg) was administered for seven days and protected distinctly hippocampal CA1 and CA3 areas against neurodegeneration. However, no effect on spontaneous seizures was observed [19]. A very similar situation occurred when valproate was tested for neuroprotection and anti-epileptogenic effect in rats with SE induced by prolonged electrical stimulation of the basal amygdala [20]. After 4 h of seizure activity, SE was stopped by diazepam and followed by valproate at an initial dose of 400 mg/kg and continued for 4 weeks at 600 mg/kg daily (3 injections of 200 mg/kg). A control group also received diazepam for termination of SE and then vehicle replaced valproate. The results indicate that valproate exerted a significant neuroprotective activity in the hippocampal area and even dentate gyrus was included. However, after the period of 4 weeks when valproate administration was stopped, no differences were shown in spontaneous seizure frequency and severity when compared to the control animals. Some benefits could be observed, though. Valproate-treated rats exhibited less hyperexcitability and locomotor hyperactivity [20]. Following kainate-induced SE in rats, valproate or phenobarbital continued to be given for 40 days at high doses and the animals were observed afterwards for the occurrence of recurrent convulsions and behavioral deficits. Furthermore, neurodegeneration was evaluated in the hippocampus [21]. Valproate distinctly reduced neurodegeneration in the areas CA1 and CA3, dentate gyrus being also protected. Rats treated with valproate displayed no spontaneous seizure activity and presented no behavioral deficits as regards emotionality and visuospatial memory. On the other hand, phenobarbital exerted no neuroprotection and did not prevent spontaneous seizures or behavioral abnormalities [21]. Phenobarbital alone was examined in another study with the use of kainate SE in immature (35-day-old) rats [22]. One day following SE, phenobarbital was initiated and continued for additional 117 days. The animals without phenobarbital evidently presented disturbances in the water maze and handling tests and had histological brain lesions as a result of kainate SE. When compared to this group, phenobarbital administered rats exhibited a similar extent of brain lesions, frequency of spontaneous convulsions and aggressiveness. Strikingly, chronic administration of phenobarbital resulted in more disturbed memory, learning and activity [22]. Valproate was also tried in rats with SE produced by pilocarpine which was terminated by a single injection of diazepam (10 mg/kg) [23]. Then, valproate was administered at 200 mg/kg and continuously infused via Alzet pumps at 600 mg/kg daily for 21 days. Three days after the infusions were stopped, the rats were observed for the incidence of spontaneous seizure activity and neurodegeneration was studied in hippocampus, olfactory cortex, thalamus and amygdala. In contrast to the previous study by Bolanos et al. [21], valproate neither affected spontaneous convulsions nor exerted significant neuroprotection. In addition to valproate, levetiracetam was applied in the same experimental approach [23]. After the initial dose of 54 mg/kg, following the termination of SE by diazepam, levetiracetam was infused for 21 days at 50, 150 and 300 mg/kg. Some neuroprotection was found for this AED at 150 and 300 mg/kg but spontaneous seizure activity was not significantly affected [23]. Similar data were reported by Santana-Gomez et al. [24]. SE was induced by lithium/pilocarpine and stopped by two intramuscular injections of diazepam (2.5 mg/kg after 2 h and 1.25 mg/kg after 10 h). Levetiracetam at 150 mg/kg was given one hour after the first injection of diazepam and then once daily for five days. Spontaneous seizures and EEG activity were monitored for eight hours a day. Spontaneous recurrent convulsions were not affected at all but significant neuroprotection was observed in the dorsal and ventral hippocampus (dentate gyrus, hilus, CA1 and CA3 areas). Lower extracellular concentration of glutamate was also found [24]. However, according to Brandt et al. [25], levetiracetam proved completely inactive when evaluated in rats after sustained electrical stimulation of the basal amygdala. The drug was administered via osmotic pumps for 8 weeks in animals without prior termination of SE and for 5 weeks in rats with termination of SE by diazepam. As already mentioned above, levetiracetam exerted no neuroprotection and did not affect spontaneous seizures. Behavioral deficit in these animals was not reduced either [25]. The only positive study with levetiracetam was brought about in rats after kainate-induced SE. The AED was infused intracerebroventricularly via osmotic mini-pumps for 25 days following SE. The mean duration of spontaneous EEG was reduced and levetiracetam also significantly decreased the number of ectopic granule cells in the hilus [26]. Some promising results were found for gabapentin which was given in immature rats in a dose of 200 mg/kg (twice daily) for one month and then at 100 mg/kg (twice daily for 10 days) following kainate-induced SE [27]. Behavioral testing (water maze and open field) started five days after gabapentin had been stopped and histological examination of the hippocampus was carried out after behavioral tests. First of all, gabapentin given to rats during brain development, significantly reduced the incidence of spontaneous convulsions and modestly protected against hippocampal damage. Moreover, gabapentin-treated rats had no deficits in learning evaluated in the water maze test and their increased activity in the open field test (post SE rats) was normalized by gabapentin administration [27]. Another AED sharing a similar to gabapentin mechanism of action, pregabalin, was tested in a lithium/pilocarpine model of SE in immature (21 days old) and adult rats [28]. The AED (50 mg/kg) was administered 20 min following pilocarpine and then the injections of pregabalin were continued for seven days. Starting from the 8th day, the AED continued to be administered at 10 mg/kg until sacrificing. Entorhinal and piriform cortices as well as hippocampus were chosen for histological evaluation. Furthermore, seizure latency to the onset of spontaneous seizure activity was recorded but in adult animals only. In adult rats, neuroprotection was evident in layers 3–4 of ventral entorhinal cortex and layer 2 of piriform cortex whilst no protection was found in the hippocampal area. In immature rats, SE produced neurodegeneration in the dentate hilus only and the degree of damage was comparable in control and pregabalin-treated rats. Although pregabalin treatment did not prevent the occurrence of spontaneous convulsions in adult rats, their latency was considerably extended in the group receiving this AED [28]. In the lithium/pilocarpine model of SE, vigabatrin (250 mg/kg) was administered 10 min after the convulsant and then continued at 100 mg/kg for 6 days [29]. The dose was increased to 250 mg/kg and injected up till the day 45. In addition, 2 h after the induction of SE, diazepam was administered at 1–2 mg/kg for 2–3 times every 4 h in order to protect the animals against SE-induced mortality. Evident neuroprotection was revealed in the field of CA3 (almost total), CA1 and moderate neuroprotective effects were noted in the hilus of the dentate gyrus. However, the extent of the neuronal damage in the entorhinal cortex was slightly enhanced when compared to animals without vigabatrin treatment. In spite of the clear cut neuroprotection in crucial for epileptogenesis brain areas, no difference in the latency to onset of spontaneous convulsions was observed. Interestingly, glutamic acid decarboxylase activity was brought to normal in comparison with the SE only group [29]. Some data on the anti-epileptogenic potential of AEDs have been derived from mice. Eslicarbazepine acetate (150 or 300 mg/kg once daily for 6 weeks) was given to mice undergoing pilocarpine-induced SE and neurodegeneration and spontaneous convulsions were studied eight weeks following SE [30]. Analysis of EEG monitoring revealed that eslicarbazepine-tretaed mice presented a significant reduction of seizure activity which was correlated with a decreased neuronal loss in the CA1 subfield of the hippocampus. Moreover, aberrant sprouting of mossy fibers into the inner molecular layer of the dentate gyrus was substantially reduced by eslicarbazepine but only at its lower dose of 150 mg/kg [30]. In kainate-induced SE in rats administration of lacosamide proved also effective in reducing spontaneous recurrent seizures and this AED exerted neuroprotective activity in the hippocampus, decreasing mossy fiber sprouting as well [31].

Apart from individual AEDs, some combinations of these drugs were evaluated against post-SE spontaneous recurrent seizures. For instance, in lithium/pilocarpine-induced SE in rats, diazepam (1.25 and 2.5 mg/kg) was injected 2 and 10 h after SE and topiramate (10, 30 or 60 mg/kg) was administered at the onset of SE and 10 h after. Whilst neuroprotective effects of this combination were observed especially in the CA1 hippocampal field, hilus and layer III/IV of the ventral entorhinal cortex, no effect was noted on the latency and severity of spontaneous seizures [32]. However, a combined treatment of levetiracetam with topiramate proved more effective in mice [33]. These AEDs were administered to mice in the latent period after SE induced by intrahippocampal kainate. Specifically, six hours after SE, topiramate (15 or 30 mg/kg) and levetiracetam (100 or 200 mg/kg) were started and continued for five days. Video and EEG monitoring for seven days was begun on 4 and 12 weeks after SE. Spontaneous focal and generalized “electroclinical” seizures were defined as behavioral convulsions with paroxysmal EEG activity. In addition, “electrographic” seizures were also considered–these were more frequent and divided into high-voltage sharp waves and hippocampal paroxysmal discharges. Histology and immunochemistry were carried out three months after SE. Multimodal brain imaging was performed 2, 7 and 37 days post-SE. When evaluated one and three months post-SE, the combined treatment of topiramate (30 mg/kg) and levetiracetam (200 mg/kg) very distinctly reduced the spontaneous electroclinical seizure frequency by 80%. Considering their incidence, convulsions were present in all control mice (*n* = 14) whilst three out of ten animals were completely protected by the drug combination which reached the level of statistical significance. In another treatment group that lost their EEG electrode assembly and was only video monitored, this result as regards spontaneous seizures was reproducible and practically almost identical. Neither the combined treatment at lower doses nor individual treatments with topiramate/levetiracetam were protective in this respect. Strikingly, almost total neuronal loss was evident in the ipsilateral hippocampal CA1 and CA3 areas and dentate hilus in not only control mice but in animals receiving the combined treatment with topiramate and levetiracetam as well. However, no anti-inflammatory effects were exerted by the combination of AEDs. The authors also tried another combined treatment, using phenobarbital and levetiracetam. However, it turned out to be completely ineffective in all evaluated aspects [33]. Results concerning conventional and newer AEDs have been summarized in Table 1 and Table 2, respectively. Combined treatments with AEDs have been shown in Table 3.

## 4. Can Non-Antiepileptic Drugs Suppress Epileptogenesis?

A very intriguing hypothesis emerged from the study by Bar-Klein et al. [34] on the anti-epileptogenic activity of losartan, an anti-hypertensive drug, blocking angiotensin II type 1 receptors. The authors used a model of acquired epilepsy which may be encountered in patients after brain infections, trauma or stroke. In this aim, an in vivo model of vascular injury was used; in short, sodium deoxycholate was applied to the rat brain surface through a craniotomy window which resulted in a distinct extravasation of albumin complexes into the brain tissue. Albumin complexes have been documented to stimulate brain TGF-β signaling, eventually leading to neuroinflammation and epileptiform activity recorded via electrocorticography. Losartan (100 mg/kg) was given on the first day of the experimental procedure and then continued up to the day 21st via drinking water (2 g/L). Evaluation of spontaneous seizure activity started 7 days after losartan had been withdrawn and was carried out for two weeks. All control animals developed spontaneous convulsions and in the losartan group only 40% of rats became epileptic. Moreover, the average number of seizures in the control group (8 per week) was substantially reduced to 2.25 seizures/week [34]. When studied in animals with SE produced by kainate, losartan was injected in a dose of 10 mg/kg for the first three days and then continued subcutaneously up to four weeks [35]. All animals were observed for the occurrence of seizure activity for 3 months. Measurements of behavioral disturbances and neurodegeneration in the hippocampus were begun 9 weeks after SE. Losartan-treated animals exhibited a significantly longer latency to the onset of seizure activity and a selective neuroprotection in the CA1 hippocampal area was observed. In other hippocampal areas (CA3, hilus of the dentate gyrus), neuronal lesions were less intense. However, losartan-treated rats performed considerably better in a battery of behavioral tests (sucrose preference test, open-field test, elevated plus-maze test, forced swimming test). Further data on this anti-hyperstensive drug exist in terms of its presumed anti-epileptogenic and neuroprotective properties are required. Basically, the experimental approach was comparable to that of the previous study [35]. The only difference concerned spontaneously hypertensive rats because the authors intended to study the influence of losartan in a model of co-existing hypertension and epilepsy, following kainate-induced SE [36]. The AT1 receptor antagonist extended the latency to the onset of spontaneous seizures, also reducing their frequency and duration. Its beneficial effects upon spontaneous seizure activity was associated with a clear cut neuroprotective activity found in the CA3 hippocampal area and the septo-temporal hilus of the dentate gyrus. Unlike the former study [35] losartan-treated animals still displayed behavioral disturbances evaluated in the four mentioned above behavioral tests [36]. Amygdala-kindled seizures in rats may be regarded as a model of epileptogenesis as a number of subconvulsive electrical stimulations eventually results in progressive seizure activity [37]. When used in this model, losartan (both following systemical or intracerebroventricular administration) retarded severe behavioral convulsions as well as EEG-recorded afterdischarges [38]. Moreover, the drug increased the latency to fully developed kindled seizures but did not modify the afterdischarge threshold or seizure severity in fully kindled rats [37]. Another non-AED may be considered for blocking epileptogenesis. Actually, isoflurane, an inhaled anesthetic agent, was given in rats during kainate-induced SE or after SE produced by paraoxon [39]. Majority of rats developed spontaneous recurrent convulsions following SE but isoflurane, although not affecting duration and severity of kainate-induced seizures, significantly reduced the number of spontaneously convulsing animals. However, isoflurane diminished the degree of neuroinflamation. As regards rats surviving paraoxon-induced SE, isoflurane also distinctly diminished the number of animals with spontaneous convulsions. Apart from this effect, isoflurane protected the blood/brain barrier from dysfunction and provided evident neuroprotection [39].

Rapamycin, an immunosuppressive drug belonging to the group of agents blocking mTOR complex 1 (mammalian target of rapamycin) pathway [40], has been also shown to possess some anti-epileptogenic potential. Similarly to the drugs mentioned above, it has been demonstrated to reduce consequences of animal SE. For instance, when administered to rats following SE produced by kainate at 10 mg/kg (subsequently terminated with sodium pentobarbital at 30 mg/kg), rapamycin (6 mg/kg every four days) significantly reduced the number of spontaneous seizures by 60% and 50% on days 17 and 21 post-SE, respectively [41]. When SE was induced by electrical stimulation of the rat angular bundle, rapamycin (at 6 mg/kg once daily for one week) was a strong inhibitor of spontaneous seizure activity, blocking it totally in 25% of rats and significantly reducing in the remaining animals [42]. Hilar cell loss and neuronal sprouting as well as blood/brain barrier permeability were potently reduced by rapamycin treatment. However, there was no difference in the expression of hippocampal inflammation markers in rapamycin-treated and control rats [42]. Negative data on the anti-epileptogenic potential of rapamycin are also available. After pilocarpine-induced SE in mice, rapamycin (1.5 or 3 mg/kg) was administered for two months. Mossy fiber sprouting was significantly reduced by rapamycin but there was no difference in spontaneous seizure frequency [43]. In pilocarpine (300 mg/kg)-induced SE in mice which was terminated with diazepam (10 mg/kg) two hours after the onset, rapamycin (10 mg/kg daily) was initiated on the next day and continued up to two months [44]. In rapamycin-treated mice, pronounced inhibition of mossy fiber sprouting was observed, however, recurrent spontaneous convulsions were not affected [44]. As already mentioned above, rapamycin is an inhibitor of mTOR complex 1 pathway so a compound blocking two cellular mTOR complexes, PQR620 (1,3,5-triazine derivative) has been evaluated in the intrahippocampal kainate model in mice [45]. The compound was given for two weeks and video/EEG recording was initiated six weeks after the treatment was discontinued. No effects of PQR620 on spontaneous seizure frequency or incidence were noted. However, granule cell dispersion in the dentate gyrus was not affected but PQR620-pretreated mice presented distinct reduction of anxiety [45]. In the same model of SE in mice, chronic rapamycin administration decreased granule cell dispersion and sprouting of the mossy fibers without affecting the total loss of neuronal cells. What is particularly important, rapamycin did not completely affect hippocampal paroxysmal discharges developing after the silent period following SE [46].

Celecoxib (a non-steroidal anti-inflammatory drug-blocker of the cyclooxygenase 2 and HMGB1/TLR-4 inflammatory pathways) represents another example of a non-AED, possessing an anti-epileptogenic potential. The drug administration (per os, 20 mg/kg) was begun one day after lithium/pilocarpine SE (terminated with diazepam at 10 mg/kg two hours after pilocarpine injection) and continued throughout the whole latent period until the day 28^th^ in rats. Histology and immunohistochemistry were carried out 14 or 28 days after SE. Spontaneous seizure activity was recorded between days 28–42 [47]. When the latent period was considered (14 and 28 days), celecoxib exerted potent neuroprotective effects in the CA1, CA3 areas and hilus of the dentate gyrus. Moreover, it inhibited microglia activation and aberrant neurogenesis/gliogenesis. Spontaneous recurrent seizures were considerably attenuated in terms of their frequency and duration [47].

## 5. Clinical Data

Clinical studies on the inhibition of epileptogenesis concern patients with brain injuries and the development of posttraumatic epilepsy in some of them. Posttraumatic epilepsy may develop after severe head injuries in about 7.1% of patients after one year and in 11.5% after five years as revealed from a population-based cohort of 2747 injured patients [48]. According to Herman [49], the risk of epilepsy in such patients with moderate/severe traumatic brain injury has been estimated in the range of even 25%–30% over two years. Some conventional AEDs were applied to such patients in order to prevent posttraumatic epilepsy. However, clinical outcomes with the use of carbamazepine, phenobarbital, phenytoin or valproate were not encouraging at all [14,50,51,52]. Of note, valproate administered for six months after a head injury even tended to increase mortality without any preventing activity against posttraumatic epilepsy [48]. Another, more effective option could be levetiracetam which was used at 55 mg/kg daily for 30 days, starting from the day 8th after the head trauma [53]. After two years, in 13.3% of patients without prophylactic treatment (*n* = 60) and in 9.1% pretreated with levetiracetam (*n* = 66) posttraumatic epilepsy was observed. Apparently, these results did not reach the level of significance. However, levetiracetam was well tolerated so only a small proportion of patients did not complete the study [53]. Nevertheless, a broad review on the possible antiepileptogenic activity of phenytoin or levetiracetam provided obviously not encouraging data because none of these AEDs could efficiently reduce the number of patients with late onset seizures [54].

## 6. Examples of New Antiepileptogenic Agents among Diverse Groups of Chemicals, Including Already Approved Drugs

No doubt, the pharmacological symptomatic treatment of epilepsy seems to be limited by the fact that around 30% of patients with epilepsy cannot be totally protected against seizure activity [1,2,3,4]. A concept that prevention or inhibition of epileptogenesis, as a causative treatment, may be more efficient, gains more and more attention. That is why probable anti-epileptogenic compounds have been searched for not only among AEDs or other approved non-epileptic drugs but also among agents affecting various pharmacological targets. Because this is not the main objective of the present review, some promising anti-epileptogenic agents will be only shortly mentioned. One of the candidates for epileptogenesis inhibition seems resveratrol (a phytoalexin-plant-derived compound) documented to inhibit development of remote seizures, neurodegeneration in the hippocampal areas CA1 and CA3a (but not in the CA3b and hilus) and mossy fiber sprouting [55]. It was also effective against pentylenetetrazol-induced kindled seizures in rats which was reflected by extending seizure latency and reducing the seizure score. Resveratrol also offered some protection against neurodegeneration and reduced seizure-related oxidative stress [56]. Cognition in pentylenetetrazol-kindled rats was also improved by this compound [57]. In the lithium/pilocarpine model of SE in rats, WIN55,212-2 (a non-selective cannabinoid agonist), however, in spite of its clear cut neuroprotective activity in the dentate hilus, it did not affect the development of spontaneous seizure activity [58]. Some initial data on the neuroprotective and anti-epileptogenic potential of cannabidiol (a non-psychostimulant phytocannabinoid; Epidiolex), however, are available. Cannabidiol, actually reduced cognitive impairment and occurrence of spontaneous seizures in animal models of status epilepticus. However, the drug has been found effective in pediatric epilepsy [59,60]. Recent data indicate that antisense oligonucleotids reducing SE-produced expression of microRNA-134 have been effective in attenuation the spontaneous seizure activity [61]. In addition to celecoxib, some other cyclooxygenase inhibitors (for instance, SC58236) were tried as anti-epileptogenic agents after electrically-induced SE through electrodes placed in the angular bundle in rats. No protection was offered against remote seizures and SE-induced neurodegeneration [62,63]. However, targeting different inflammatory pathways by binary combinations of anti-inflammatory agents proved effective against remote consequences of lithium/pilocarpine SE in rat pups. Clear cut neuroprotection was observed and the animals were protected against spontaneous convulsions [64]. Apart from a number of drugs or agents, also ketogenic diet proved anti-epileptogenic in rats after pilocarpine-induced SE. Specifically, rats on a ketogenic diet expressed a reduced number of spontaneous convulsions [65].

## 7. Conclusions

AEDs clearly represent a symptomatic approach in the treatment of epilepsy, reducing the frequency and severity of seizures [66]. It has been widely documented that pilocarpine-induced SE in rats is causally associated with diffuse seizure-dependent brain damage [67] and as indicated above, neurodegeneration seems closely related with epileptogenesis. Therefore, a question emerged whether AEDs presenting neuroprotective potential against SE-induced neurodegeneration would be also capable of epileptogenesis inhibition? The data derived from the experiments on the inhibition by AEDs of SE-induced neurodegeneration and spontaneous seizures (emerging after a seizure-free period) are equivocal. First, there are antiepileptics providing neither neuroprotection nor inhibition of spontaneous seizure activity (levetiracetam, phenobarbital, valproate) [18,19,20,21,22]. Second, there is a group of AEDs exerting neuroprotective effects in the hippocampus or dentate gyrus and yet without influence on spontaneous convulsions (levetiracetam, topiramate, valproate, vigabatrin) [18,23,24,25,27]. Some combinations of AEDs proved also neuroprotective (topiramate + diazepam) but not preventing spontaneous seizure activity and some (topiramate + phenobarbital) completely ineffective [32,33]. A number of AEDs exerted both neuroprotection and reduced the number of spontaneous convulsions (diazepam, eslicarbazepine, gabapentin, lacosamide, pregabalin) [17,30,31]. Unexpectedly and surprisingly, some AEDs combined (levetiracetam + topiramate) were capable of reducing spontaneous seizures without any neuroprotective potential [33]. There are also examples of antiepileptics exerting neuroprotection and reducing the behavioral deficit in epileptic animals although without any protective effect upon spontaneous seizures (valproate) [20].

Obviously, phenobarbital, vigabatrin and levetiracetam should not be considered as anti-epileptogenic drugs. Out of three studies on levetiracetam, all of them reported no inhibition of spontaneous seizure activity in animals surviving SE [23,24,25] and only one pointed to its neuroprotective activity [24]. Nevertheless, in one study levetiracetam reduced the duration of spontaneous EEG seizures which was associated with a decrease in ectopic granular cells [26]. Eslicarbazepine, gabapentin, lacosamide and pregabalin provided neuroprotection and efficiently reduced spontaneous seizures, gabapentin even improving behavioral deficit [27,28,30,31]. However, these AEDs require more evaluations in other models of SE before the strong preclinical recommendation. Valproate was tested by three research teams. According to Klitgaard et al. [23], the drug possessed neither neuroprotective nor spontaneous seizure preventive potential in pilocarpine-induced SE. Brandt et al. [20] showed its neuroprotective effect without any influence on spontaneous seizure activity. Moreover, valproate distinctly reduced the behavioral deficit in rats (less hyperexcitability and locomotor hyperactivity) in animals surviving SE produced by electrical stimulation of the amygdala. Full positive profile of this AED was found by Bolanos et al. [21] who reported neuroprotective and spontaneous seizure inhibiting properties of valproate along with better behavioral performance of rats after kainate-induced SE. These examples clearly illustrate completely different results concerning the AED which in all three studies was administered in high effective doses for a long period of time. However, the best outcome was observed after the longest period of administration reaching 40 days [21]. It is also important to recall that, however, diazepam at a single dose of 20 mg/kg was capable of significant inhibition of epileptogenesis [18]. Certainly, there are also other variables which could be of importance —for instance, a method of inducing SE which was different in each case. This underlines a necessity for a much broader evaluation of other AEDs already mentioned above eslicarbazepine, gabapentin, lacosamide and pregabalin.

One of the most prominent mechanisms of epileptogenesis is mossy fiber sprouting (see above). A very exciting hypothesis may be thus given that AEDs via blockade of this process may inhibit epileptogenesis. Some experimental data on the effects of lacosamide [31] or eslicarbazepine [30] seem to be in line with this hypothesis. If that assumption were to be true, it would have massive repercussions in our understanding of the end points for the search of anti-epileptogenic strategies. However, already mentioned above studies on rapamycin, as a putative anti-epileptogenic compound, have revealed that this drug potently inhibited mossy fiber sprouting and yet did not inhibit epileptogenesis [43,44,46].

Strikingly, the most effective in inhibiting spontaneous seizure activity was a combination of levetiracetam with topiramate which exhibited no neuroprotective or anti-inflammatory potential and efficiently inhibited the frequency and incidence of spontaneous seizure activity in mice after intrahippocampal kainate-induced SE [33]. This is particularly striking in terms of growing evidence suggesting a significant role for various cells of the innate system of the central nervous system in epileptogenesis. Actually, infiltration by monocytes and activation of microglia and perivascular macrophages were considerably increased in both human and rat epileptogenic hippocampus, in patients who did not survive status epilepticus or in rats after electrically-induced SE, respectively [68]. Anyway, in this case anti-inflammatory mechanisms seem, unexpectedly, to play no role although, as indicated above, some anti-inflammatory drugs exhibited anti-epileptogenic activity. Whether this effect could be entirely ascribed to the inhibited inflammation remains disputable.

Preclinical evaluation of various combinations of AEDs has been performed and a number of supra-additive combinations has been delineated with isobolography [1,69]. Interestingly, the combined treatment of levetiracetam + topiramate in many fixed dose ratios was supra-additive as evaluated in the mouse maximal electroshock model. Moreover, this combination did not affect either long-term memory or motor coordination [70]. Considering the mechanisms of action of these AEDs, they interact with many channels, receptors and proteins and this may explain their particular efficacy in terms of anticonvulsant and anti-epileptogenic activity when given in combination. Another combination of levetiracetam with phenobarbital tested by Schidlitzki et al. [33] was, however, completely ineffective as regards neuroprotection and spontaneous convulsions. Of note, this combination was simply additive when evaluated with isobolography [70]. The ongoing preclinical studies for the search of effective anti-epileptogenic combinations of AEDs (or other agents) have to be carried out with the use of different models of SE and in animals after brain injury. Possibly drug combinations targeting a number of mechanisms involved in epileptogenesis could be more effective. Any future preventive treatment in patients with a high risk of epilepsy (for instance after SE or stroke) have to be correlated with possible markers for epileptogenesis [6]. On one hand, such a marker could identify patients endangered with epilepsy and on the other, it could be of value in monitoring treatment response. So far, one of the probable candidates is translocator protein 18 kDa (TSPO) which via molecular imaging may give insight into microglia activation during epileptogenesis [6]. Some other potential markers (metabolites, proteins, mRNAs and miRNAs being altered during epileptogenesis) have to be also considered [71].

Clinical data on the preventive use of AEDs in patients after head traumas are not encouraging and as indicated above valproate even showed a tendency to increase mortality in these patients. A possibility exists that combinations of AEDs with already established anti-epileptogenic potential could be effective in this regard. No data exist on any attempts to inhibit epileptogenesis in clinical trials due to the lack of reliable markers for epileptogenesis. Some hope may emerge from optogenetics which enables controlling or recording neural activities with light following delivery into the brain of light-sensitive opsin genes [72]. This approach seems effective not only in seizure inhibition in animal models [72] but in alleviating epileptogenesis as well in rats with opsin genes in dentate gyrus of the hippocampus [73].

## Figures and Tables

**Table 1 ijms-21-02340-t001:** Conventional antiepileptic drugs and their effects of status epilepticus-induced neurodegeneration, spontaneous seizure activity and behavioral deficit in rats.

Antiepileptic Drug	Status Epilepticus	Neuroprotection	Spontaneous Seizures	Behavioral Deficit
Carbamazepine (14 days)	Electrical stimulation of perforant path	Not evaluated	Not evaluated	Present [16]
Diazepam (single injection in a high dose)	Electrical stimulation of amygdala	Present	Reduced	Not evaluated [17]
Diazepam (single injection in a low dose)	Lithium/pilocarpi- ne	Moderate	Not affected	Not evaluated [18]
Phenobarbital (in immature rats for 117 days)	Kainate	None	Not affected	Enhanced [22]
Phenobarbital (for 40 days)	Kainate	None	Not affected	Present [21]
Valproate (for 28 days)	Electrical stimulation of amygdala	Present	Not affected	None [20]
Valproate (for 21 days)	Pilocarpine	None	Not affected	Not evaluated [21]
Valproate (for 40 days)	Kainate	Present	Reduced	None [20]

Antiepileptic and convulsant drugs were administered intraperitoneally.

**Table 2 ijms-21-02340-t002:** Newer antiepileptic drugs and their influence on status epilepticus-induced neurodegeneration, spontaneous convulsions and behavioral deficit in rodents.

Antiepileptic Drug	Status Epilepticus	Neurodegeneration	Spontaneous Convulsions	Behavioral Deficit
Eslicarbazepine (in mice for 42 days)	Pilocarpine	Present with mossy fiber sprouting reduced	Reduced	Not evaluated [30]
Gabapentin (for 40 days)	Kainate	Present	Reduced	Reduced [27]
Lacosamide	Kainate	Present with mossy fiber sprouting reduced	Reduced	Not evaluated [31]
Lamotrigine (for 14 days)	Electrical stimulation of performant path	Moderate	Not evaluated	Present [16]
Levetiracetam (for 5 days)	Lithium/pilocar-pine	Present	Not affected	Not evaluated [24]
Levetiracetam (for 35 or 56 days via osmotic pumps)	Electrical stimulation of amygdala	None	Not affected	Present [25]
Levetiracetam (for 21 days)	Pilocarpine	Moderate	Not affected	Not evaluated [23]
Levetiracetam (for 25 days) intracerebroventicularly via osmotic minipumps)	Kainate	Reduced number of ectopic granule cells	Reduced	Not evaluated [26]
Pregabalin (for 7 days)	Lithium/pilocar-pine	Present in entorhinal and piriform cortex	Only latency extension	Not evaluated [28]
Topiramate (for 6 days)	Lithium/pilocar-pine	Moderate (CA1 and CA3)	Not affected	Not evaluated [18]
Topiramate (for 7 days)	Lithium/pilocar-pine	Potent in CA1, moderate in CA3	Not affected	Not evaluated [19]
Vigabatrin (for 45 days)	Lithium/pilocar-pine	Pronounced in CA1 and CA3, moderate in the hilus	Not affected	Not evaluated [29]

All experiments were performed in rats, antiepileptic drugs and convulsants being administered peripherally, unless otherwise stated.

**Table 3 ijms-21-02340-t003:** Influence of combinations of antiepileptic drugs on neurodegeneration and spontaneous seizure activity produced by status epilepticus in rodents.

Antiepileptic Drugs	Status Epilepticus	Neurodegeneration	Spontaneous Seizures
Topiramate + diazepam (for 7 days in rats)	Lithium/pilocarpine intraperitoneally	Partial in CA1, hilus and entorhinal cortex	Not affected [31]
Levetiracetam + topiramate (for 5 days in mice)	Intrahippocampal kainate	Present	Reduced [32]
Levetiracetam + phenobarbital (for 5 days in mice)	Intrahippocampal kainate	Present	Not affected [32]

Antiepileptic drugs were administered intraperitoneally.

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
