# Peer review of "Anti-Epileptogenic Effects of Antiepileptic Drugs"

_ijms, 2020, doi:10.3390/ijms21072340_

Round 1
Reviewer 1 Report
The authors would like to show anti-epileptogenic effects of AEDs.
The contents are correct and informative to readers.
I have 2 concerns on the title.
- The review article is on epileptogenesis/anti-epileptogenesis of AEDs. It is completely different from neuroprotection/neuroprotective effects. So please kindly re-consider the correct title.
- The AEDs have several neuroprotective effects. So if the authors would like to show 'neuroprotective effects of AEDs, please kindly add more information on neuroprotection and the mechanisms.
Author Response
Answers to the referees’ comments and suggestions (MS# ijms-759514)
Reviewer 1.
- The review article is on epileptogenesis/anti-epileptogenesis of AEDs. It is completely different from neuroprotection/neuroprotective effects. So please kindly re-consider the correct title.
We quite agree with this remark. The title was changed accordingly and now it reads: “Anti-epileptogenic effects of antiepileptic drugs?”
- The AEDs have several neuroprotective effects. So if the authors would like to show 'neuroprotective effects of AEDs, please kindly add more information on neuroprotection and the mechanisms.
A short text (p. 6, 2nd para, l. 4-8) was added on this issue, just to make sure the readers which mechanisms may come into question.
Reviewer 2 Report
This is an excellent and thorough review of the topic. The tables are very helpful in summarizing the information.
As this is a review of neuroprotective effects of antiepileptic drugs in epilepsy, I would also briefly comment on the neuroprotective findings of CBD and the ketogenic diet in epilepsy or refer to them.
There are good descriptions regarding structural changes,such as in the hippocampus with mossy fiber sprouting, etc. I would also comment on inflammatory changes.
THere is growing evidence various pathways link the immune response and seizures. These include adaptive systemic responses, such as T- and B-cell activation and auto-AB production, and innate mechanisms of the CNS, like the increased production of cytokines by activated glial cells observed in response to various stimuli such as seizures. These can lead to neuronal functional changes and increased epileptogenesis. Any studies looking at antiepileptics agents and neuroprotection in this setting?
The Conclusions section is too long. I'd recommend making it more succinct to summarize the findings.
Author Response
Answers to the referees’ comments and suggestions (MS# ijms-759514)
Reviewer 2.
- As this is a review of neuroprotective effects of antiepileptic drugs in epilepsy, I would also briefly comment on the neuroprotective findings of CBD and the ketogenic diet in epilepsy or refer to them.
Short comments on CBD and ketogenic diet were added. The first one may be found on p. 19 (l. 7-11) and the second one – p. 19 (l. 4-6 from bottom).
- There are good descriptions regarding structural changes, such as in the hippocampus with mossy fiber sprouting, etc. I would also comment on inflammatory changes. There is growing evidence various pathways link the immune response and seizures. These include adaptive systemic responses, such as T- and B-cell activation and auto-AB production, and innate mechanisms of the CNS, like the increased production of cytokines by activated glial cells observed in response to various stimuli such as seizures. These can lead to neuronal functional changes and increased epileptogenesis. Any studies looking at antiepileptic agents and neuroprotection in this setting?
Although there is no doubt that inflammatory processes participate in epileptogenesis and some anti-inflammatory drugs possess some anti-epileptogenic activity (especially when combined), the combination of topiramate with levetiracetam, exhibiting no anti-inflammatory potential, seems to contradict the involvement of anti-inflammatory mechanisms in anti-epileptogenic activity. Now, this is shortly referred to on p. 22, l. 1-7 from bottom – 23. l.1-3.
- The Conclusions section is too long. I'd recommend making it more succinct to summarize the findings.
We did our best to condense this section and the listed texts below have been deleted:
Also two publications on phenobarbital underlie no protective potential of this AED upon neuronal lesions and spontaneous convulsions [20, 21]. Vigabatrin, in spite of its neuroprotection also did not inhibit spontaneous seizure activity [28].
In the other two studies, these periods were 21 [22] and 28 days [19]. One cannot thus exclude a possibility that the length of valproate’s administration may be a crucial factor determining its efficacy
According to Buckmaster and Lew [41], rapamycin did not influence proliferation of granule cells, loss of hilar neurons or generation of ectopic granule cells. Perhaps these cellular targets will be better correlated with the inhibition of epileptogenesis. A study documenting the anti-epileptogenic potential and reduction of neuronal sprouting by rapamycin, however, also exists [41]. Furthermore, levetiracetam was shown to reduce the number of ectopic granule cells and decrease the duration of spontaneous EEG seizures, but as indicated above, was totally ineffective in other three models of SE.
There are some other supraadditive combinations of AEDs with encouraging profile of neurotoxicity delineated in preclinical studies – for instance, topiramate +lamotrigine, lamotrigine + valproate or levetiracetam + gabapentin [1,65]. At least one of them may possess a clear cut antiepileptogenic potential.
To be true, some new paras have been added so the net reduction may seem small but we do hope that the referee is able to accept it as the Conclusions section in our opinion is of key importance.
Reviewer 3 Report
At the manuscript "Neuroprotective effects of antiepileptic drugs in epilepsy" by Dr. Barbara Miziak et al authors reviewed materials answering for the question “are antiepileptic drugs able to inhibit status epilepticus induced neurodegeneration? and can a neuroprotective potential of particular antiepileptic drug stop epileptogenesis?
Clinical and experimental data shown that only particular drugs exhibited potent or moderate neuroprotective effects in animal models. The combination of two drugs - levetiracetam and topiramate proved highly synergistic in different models of the epileptic seizures; moreover, this combination inhibited epileptogenesis in rats following status epilepticus. Authors concluded that there is no correlation between neuroprotection and antiepileptogenetic effect of this combination. The presentation of a subject is systematic and comprehensive. List of references is quite full and statistical analysis is proper.
My notes are only of a minor nature:
1. The manuscript that the authors cite (Schidlitzki et al, Neurobiol. Dis. 2020, 134, 104664) uses two modalities of brain imaging – MRI and PET. Comparison of different imaging methods always provides a very valuable information output. So I strongly advise you to add also optical and some other imaging of the epileptic seizures: Advantages and limitations of brain imaging methods in the research of absence epilepsy in humans and animal models:
Bortel et al, Epilepsy Res. 2019 Nov;157:106209.
Lenkov et al, J Neurosci Methods. 2013 Jan 30;212(2):195-20
Zhao et al, Epilepsy Res. 2015 Oct;116:15-26.
2. Authors describing pharmacological mechanism of levetiracetam. Neural networks changes associated with levetiracetam treatment has been recently described:
Pang et al, Neurol Sci. 2020 Mar 9. doi: 10.1007/s10072-020-04322-8.
Perhaps authors can cite and use this paper.
3. As authors wrote, some protein can be used for molecular imaging to clarify role of microglia activation during epileptogenesis. More data about such methods can be find in:
Dixit AB, Tripathi M, Chandra PS, Banerjee J. Molecular biomarkers in drug-resistant epilepsy: Facts & possibilities. Int J Surg. 2016;36(Pt B):483-491
After corrections, mentioned above, I will be happy to recommend the manuscript for publication.
Author Response
Answers to the referees’ comments and suggestions (MS# ijms-759514)
Reviewer 3.
My notes are only of a minor nature:
- The manuscript that the authors cite (Schidlitzki et al, Neurobiol. Dis. 2020, 134, 104664) uses two modalities of brain imaging – MRI and PET. Comparison of different imaging methods always provides a very valuable information output. So I strongly advise you to add also optical and some other imaging of the epileptic seizures: Advantages and limitations of brain imaging methods in the research of absence epilepsy in humans and animal models:
Bortel et al, Epilepsy Res. 2019 Nov;157:106209. Lenkov et al, J Neurosci Methods. 2013 Jan 30;212(2):195-20 Zhao et al, Epilepsy Res. 2015 Oct;116:15-26.
In our opinion, the paper by Zhao et al. (2015) adds an intriguing possibility to counteract epilepsy and in the future – epileptogenesis. Our comments were added at the end of the Conclusions section (p. 23, last para).
- Authors describing pharmacological mechanism of levetiracetam. Neural networks changes associated with levetiracetam treatment has been recently described:
Pang et al, Neurol Sci. 2020 Mar 9. doi: 10.1007/s10072-020-04322-8.
Perhaps authors can cite and use this paper.
This novel mechanism of levetiracetam was considered and a relevant comment may be found on p. 3 (l. 1-2 from bottom) – 4 (l. 1-2).
- As authors wrote, some protein can be used for molecular imaging to clarify role of microglia activation during epileptogenesis. More data about such methods can be find in:
Dixit AB, Tripathi M, Chandra PS, Banerjee J. Molecular biomarkers in drug-resistant epilepsy: Facts & possibilities. Int J Surg. 2016;36(Pt B):483-491.
Basing on this paper, some comments on possible markers for epileptogenesis were made at the end of the last section (p. 23, end of the 2nd para).
For the convenience of the referees, all corrections have been highlighted in yellow.
The authors deeply appreciate the constructive referees’ comments which allowed to perform some adequate amendments.